# Piceatannol Ameliorates Hepatic Oxidative Damage and Mitochondrial Dysfunction of Weaned Piglets Challenged with Diquat

**DOI:** 10.3390/ani10071239

**Published:** 2020-07-21

**Authors:** Peilu Jia, Shuli Ji, Hao Zhang, Yanan Chen, Tian Wang

**Affiliations:** College of Animal Science and Technology, Nanjing Agricultural University, Nanjing, Jiangsu 210095, China; 2018105059@njau.edu.cn (P.J.); 2018105058@njau.edu.cn (S.J.); h.zhang@njau.edu.cn (H.Z.); 2019205019@njau.edu.cn (Y.C.)

**Keywords:** piceatannol, diquat, oxidative stress, mitochondrial function, apoptosis, piglet

## Abstract

**Simple Summary:**

In our experiment, piglets in two challenged groups were orally administrated either piceatannol or another vehicle solution, and then injected with diquat, a bipyridyl herbicide that can cause a large amount of reactive oxygen species in animal bodies and is widely used to cause oxidative stress, to investigate the effects of piceatannol on hepatic redox status, mitochondrial function and the underlying mechanism. A control group was given neither piceatannol supplementation nor diquat injection. Results showed that piceatannol could improve hepatic redox status, preserve mitochondrial function, and prevent excessive apoptosis of liver cells. In addition, piceatannol might exert its protective effects through a classic antioxidant signaling pathway named Nrf2. Our findings indicated that piceatannol might be an appropriate candidate for further development as an antioxidant food supplement to minimize the risk of oxidative stress in young animals.

**Abstract:**

The liver is an organ that produces large amounts of reactive oxygen species (ROS). Human infants or piglets are prone to oxidative damage due to their uncompleted development of the antioxidant system, causing liver disease. Piceatannol (PIC) has been found to have significant antioxidant effects. The aim of this experiment was to investigate the effects of PIC on the liver in piglets experiencing oxidative stress caused by diquat (DQ). After weaning, 54 male piglets (Duroc × [Landrace × Yorkshire]) were selected and randomly divided into three treatment groups: the CON group, the DQ-CON group, and the DQ-PIC group. The two challenged groups were injected with DQ and then orally administrated either PIC or another vehicle solution, while the control group was given sterile saline injections and an orally administrated vehicle solution. Compared to the results of the CON group, DQ increased the percentage of apoptosis cells in the liver, also decreased the amount of reduced glutathione (GSH) and increased the concentration of malondialdehyde (MDA). In addition, the adenosine triphosphate (ATP) production, activities of mitochondrial complex I, II, III, and V, and the protein expression level of sirtuin 1 (SIRT1) were inhibited by DQ. Furthermore, PIC supplementation inhibited the apoptosis of hepatic cells caused by DQ. PIC also decreased MDA levels and increased the amount of GSH. Piglets given PIC supplementation exhibited increased activities of mitochondrial complex I, II, III, and V, the protein expression level of SIRT1, and the ATP production in the liver. In conclusion, PIC affected the liver of piglets by improving redox status, preserving mitochondrial function, and preventing excessive apoptosis.

## 1. Introduction

Oxidative stress is a mechanism commonly implicated in the pathogenesis and progression of liver diseases [1]. Oxidative stress occurs when the production of reactive oxygen species (ROS) exceeds the cellular antioxidant capacity, leading to oxidative damage to the liver that causes cell apoptosis, which leads to chronic liver disease (CLD) [2]. As a crucial organ that regulates nutrient metabolism, detoxification, and immunity, the liver is vulnerable to oxidative damage because of its various ROS sources. Kupffer cells are abundant in the liver and can produce a large amount of active oxygen while exerting a bactericidal effect. When the antioxidant activity of the Kupffer cells is insufficient, damage and apoptosis of healthy parenchymal cells occur, causing liver damage [2]. Additionally, the large number of mitochondria in the liver play an important role in the metabolic function of the liver, and these mitochondria are both the source of ROS and a target vulnerable to ROS attacks, making the liver sensitive to oxidative stress damage [3].

In order to counterbalance the ROS, cells are equipped with multiple cellular antioxidant defenses, including glutathione (GSH), superoxide dismutase (SOD) and catalase (CAT), which act to neutralize ROS or repair the damages caused by ROS [4]. However, young animals that have not yet fully developed an antioxidant system can also accumulate free radicals and propagate lipid peroxidation [5]. This raises the possibility that the use of natural antioxidants may represent promising therapeutic treatments for hepatic oxidative stress and the associated damage.

Previous studies have demonstrated that plant polyphenols are capable of scavenging free radicals and ameliorating liver diseases associated with oxidative stress. Piceatannol (PIC), a beneficial compound primarily found in grapes, rhubarb, and sugarcane, has attracted much attention in recent years due to its excellent antioxidant activity. The antioxidant property in PIC comes from the ethylene double bonds and phenolic hydroxyl groups [6,7] and can directly remove superoxide anion free radicals. PIC also has been found to alleviate lipopolysaccharide (LPS)‒induced oxidative stress in the liver of mice [8]. In addition, PIC may up-regulate the expression of heme oxygenase 1 (*HO1*) and superoxide dismutase (*SOD1*), which are downstream genes of nuclear-factor-erythroid-2-related factor 2 (*Nrf2*), in C2C12 myotubes [9] and protect ARPE-19 cells against vitamin A dimer-mediate oxidative stress via signaling [10], indicating that PIC might exert its bioactivities via the Nrf2 pathway. However, the information on the effect and potential mechanism of PIC intervention in liver damage is insufficient.

Diquat (DQ) is widely regarded as an effective chemical substance that induces oxidative stress [11], and its main target organ is the liver [12]. DQ is a common contact bipyridyl herbicide, which uses molecular oxygen to produce superoxide anion free radicals [13] or to directly attack mitochondria, resulting in mitochondrial damage and increased production of ROS [14]. Substantial evidence exists establishing the model of oxidative stress induced by diquat [15,16]. Because its digestive system is anatomically and functionally similar to human infants, piglet models have been widely used as a representative in nutrition research [17]. Thus, the present study was conducted using a porcine model to explore the potential of PIC to alleviate diquat-induced liver injury and to illustrate the detailed mechanisms of its antioxidant effect. We supposed that PIC can inhibit the degree of cell apoptosis in liver, reduce the degree of liver oxidative damage via the Nrf2 signaling pathway and have protective effects on mitochondrial function and energy metabolism.

## 2. Materials and Methods

### 2.1. Ethical Statement

This experiment complied with Laboratory Animals Standards of Welfare and Ethics (DB32/T 2911-2016), and all experimental procedures involved were approved by the Institutional Animal Care and Use Committee of Nanjing Agricultural University (Permit number SYXK-2017-0027l; Nanjing, China).

### 2.2. Animals and Treatments

A total of 54 male piglets (Duroc × (Landrace × Yorkshire)) with an average body weight (BW) of 8.13 ± 0.41 kg were selected at 28 days of age and then randomly divided into three treatment groups; each treatment group contained six replicates with three piglets per replicate (*n* = 6): (1) the CON group (CON), in which the piglets were orally administered a vehicle solution (0.5% sodium carboxymethyl cellulose, Sigma-Aldrich Corp., St. Louis, MO, USA) from 28 to 35 days of age and were challenged with sterile saline at 35 days of age; (2) the DQ-CON group (DQ-CON), in which the piglets were orally administered a vehicle solution from 28 to 35 days of age and challenged with DQ [18,19,20,21] (10 mg/kg body weight, Sigma-Aldrich Corp., St. Louis, MO, USA) at 35 days of age; and 3) the DQ-PIC group (DQ-PIC), in which the piglets were orally administrated PIC (80 mg/kg/day, Great Forest Biomedical Ltd., Hangzhou, China) from 28 to 35 days of age and challenged with DQ (10 mg/kg body weight) at 35 days of age. The dosage of piceatannol (80 mg/kg/d) on piglets has not been reported in previous studies. However, the dose of piceatannol that we used in this experiment is based on our team’s research on resveratrol [22,23,24]. Piceatannol is similar to resveratrol and is a plant polyphenolic active substance [25]. As an analog of resveratrol, it has a similar structure and biological activity as resveratrol [6]. Therefore, we believe that the dose of resveratrol in weaned piglets has reference significance for PIC. Piglets were given free access to water and food during the trial; the nutrient composition of their diet is shown in Table 1. The body weight and food intake of the piglets in each replicate were recorded carefully during the feeding period to calculate the average daily gain (ADG), average daily feed intake (ADFI), feed conversion ratio (FCR) at each age between 28 to 35 days, and the change in body weight during the 24 h post-injection.

### 2.3. Sample Collection

On day 36, blood samples were obtained from the neck vein of each piglet and centrifuged at 4000 rpm for 10 min at 4 °C. Piglets were killed by exsanguination after electrical stunning. Liver samples (~5 g) taken from the same location in the liver of each piglet were transferred to ice-cold phosphate-buffered saline (PBS, pH 7.4). Additional liver samples for further analysis were stored at −80 °C after quick freezing with liquid nitrogen.

### 2.4. Evaluation of Hepatic Damage

The liver samples were fixed in 4% paraformaldehyde for 24 h then dehydrated using graded ethanol; they were then embedded in paraffin and cut into 5 micron-thick sections with a microtome. Next, the sections were baked at 42 °C for 8 h, stained with hematoxylin and eosin, and then imaged and observed under an inverted optical microscope. Terminal deoxyribonucleotidyl transferase-mediated deoxyuridine triphosphate nick end labeling (TUNEL) assay was performed using a TUNEL BrightGreen Apoptosis Detection Kit (Vazyme Biotech Co., Ltd., Nanjing, China) following the guidelines of the manufacturer. The paraffin sections were dewaxed twice with xylene, the residual xylene was removed using graded alcohol, and then the sections were incubated with proteinase-K (20 µg/mL) for 20 min. After a further incubation with the equilibration buffer for 20 min, sections were incubated with TdT in a humidified and dark incubator at 37 °C for 60 min. Sections were counterstained with DAPI (2 µg/mL) for 5 min and viewed using a fluorescence microscope (Olympus AX70TRF, Tokyo, Japan). After each step above was completed, the sections were washed in three changes of PBS. The percentage of apoptotic cells (green cells) was calculated using Image-Pro Plus (ver. 6.0 for Windows, Media Cybernetics, Inc., Rockville, MD, USA) to count at least 500 cells in 5 random fields per section using the formula (TUNEL-positive cells / total cells) × 100%.

The amount of the plasma alanine aminotransferase (ALT) and aspartate aminotransferase (AST) were measured using commercial kits (Nanjing Jiancheng Bioengineering Institute, Nanjing, China).

### 2.5. Analysis of Hepatic Redox Status

Approximately 300 mg of frozen liver samples was placed in a 1:9 (wt/vol) saline solution and then homogenized with a high-speed benchtop homogenizer (Tekmar, Cincinnati, OH, USA). The supernatant was collected by centrifugation and then used for the analysis of superoxide dismutase (SOD), glutathione peroxidase (GPx), and amount of reduced glutathione (GSH) and malondialdehyde (MDA) using commercial kits (Nanjing Jiancheng Bioengineering Institute, Nanjing, China). The total protein concentration in the homogenate was measured using a total protein assay kit (Nanjing Jiancheng Bioengineering Institute, Nanjing, China).

### 2.6. Extraction and Assessment Function of Hepatic Mitochondria

Hepatic mitochondria were extracted using a commercial mitochondrial extraction kit (Beijing Solarbio Science and Technology Co., Ltd., Beijing, China) following the manufacturer’s instructions. The activities of the hepatic mitochondrial complex and citrate synthase (CS) and the amount of nicotinamide adenine dinucleotide (NAD^+^), reduce nicotinamide adenine dinucleotide (NADH), and adenosine triphosphate (ATP) were measured with commercial kits (Beijing Solarbio Science and Technology Co., Ltd., Beijing, China) according to the manufacturer’s instructions. The total protein concentration in each sample was measured using a total protein assay kit (Nanjing Jiancheng Bioengineering Institute, Nanjing, China).

### 2.7. Determination of Hepatic Mitochondrial DNA (mtDNA) Content

The measurement of the relative mtDNA content was completed using the nuclear-encoded β-actin as an internal reference and by coamplifying the mitochondrial D-loop using a quantitative real-time polymerase chain reaction. The isolation of the total mtDNA from the liver samples was completed using a DNAiso Reagent (TaKaRa Biotechnology, Dalian, China). The quantitative real-time polymerase chain reaction was completed using a fluorescence quantitative polymerase chain reaction instrument (Applied Biosystems, Foster City, CA, USA) with the following steps: 95 °C for 10 s, 50 cycles of 95 °C for 5 s, 60 °C for 25 s, and 95 °C for 10 s. The relative mtDNA amount was calculated using the 2^−ΔΔCt^ method. The primers used in this experiment are listed in Table 2.

### 2.8. Isolation of Total mRNA and Analysis of the Quantitative REAL-Time Polymerase Chain Reaction (qRT-PCR)

Extraction of the total mRNA from the liver tissues used Trizol (Thermo Fisher Scientific Inc., Waltham, MA, USA) reagent at the recommendation of the manufacturer of the test. The light absorption of the total mRNA was determined to be 260 nm (A260) and 280 nm (A280) using a Nanodrop ND-2000c spectrophotometer (Thermo Scientific, Camden, NJ, USA). A reverse transcription of the total RNA was completed using the PrimeScript^TM^ RT Master Mix (TaKaRa, Biotechnology, Dalian, China) following the manufacturer’s guidelines. The qRT-PCR was completed using the SYBR Premix Ex Taq II kit (TaKaRa, Biotechnology, Dalian, China) and a fluorescence quantitative polymerase chain reaction instrument (Applied Biosystems, Foster City, CA, USA) in the following steps: 95 °C for 30 s, 40 cycles of 95 °C for 5 s, and 60 °C for 30 s. The primers used in this experiment are listed in Table 2. The relative expression abundance of the target genes of the mRNA was calculated using β-actin as an internal reference and expressed using the 2^−ΔΔCt^ method.

### 2.9. Isolation of Hepatic Protein and Western Blotting

The liver tissues were lysed in an ice-cold radioimmunoprecipitation assay reagent (Beyotime Biotechnology, Shanghai, China) containing protease inhibitors (Beyotime Biotechnology, Shanghai, China). Supernatant was collected after centrifugation at 12,000 *g* for 5 min and mixed with a sodium dodecyl sulfate-polyacrylamide gel electrophoresis (SDS-PAGE) sample loading buffer (Beyotime Biotechnology, Shanghai, China). Before performing gel electrophoresis, the mixture was denatured at 99 °C for 15 min.

Extracted proteins were separated using SDS-PAGE gel electrophoresis and then transferred onto the polyvinylidene fluoride (PVDF) membrane. Tris-buffered saline containing tween (TBST) was used to wash the membranes three times. Next, the membranes were blocked at room temperature for 2 h using 5% skimmed milk. After washing with TBST, the membranes were incubated with the following primary antibodies overnight at 4 °C: nuclear-factor-erythroid-2-related factor 2 (Nrf2), (Proteintech Group, Inc., Wuhan, China), kelch like ECH associated protein 1 (Keap1), superoxide dismutase 2 (SOD2), sirtuin 1 (SIRT1), B-cell lymphoma-2 (Bcl-2), and Bcl2-associated x (Bax). Thereafter, the membranes were washed in three changes of TBST and incubated with the appropriate secondary antibody for 2 h. Images of the membranes were taken with a luminescence image analyzer LAS-4000 system (Fuji Corporation, Tokyo, Japan), and quantitative analysis was performed using Image-Pro Plus (ver. 6.0 for Windows, Media Cybernetics, Inc., Rockville, MD, USA).

### 2.10. Statistical Analysis

Data are presented here as mean ± standard error (SE). Statistical Product and Service Solutions (SPSS, ver. 20.0 for Windows, SPSS Inc., Chicago, IL, USA) software was used for statistical analysis, and GraphPad Prism 7 (ver. 7.0 for Windows, GraphPad software, Inc., San Diego, CA, USA) was used for drawing columns. Differences between treatments were analyzed for one-way analysis of variance (ANOVA) using Tukey’s post hoc test for pair comparisons, which considers statistical significance to occur when the *p*-value is less than 0.05.

## 3. Results

### 3.1. Growth Performance

No difference was observed between the groups for the ADG, ADFI, or FCR of piglets during the first week after weaning (*p* > 0.05, Table 3). After the DQ challenge, the DQ group had a significant decrease in body weight when compared with the CON group (*p* < 0.05). By contrast, the oral administration of PIC restored the DQ-induced decrease in body weight (*p* < 0.05).

### 3.2. Effects of PIC on Hepatic Injury Caused by DQ in Piglets

After injecting with DQ, the amount of plasma AST and ALT of the DQ-CON group piglets increased significantly compared to the CON-group piglets (*p* < 0.05, Table 4). However, the DQ-PIC piglets had lower amounts of plasma AST and ALT than those in the DQ-CON group (*p* < 0.05).

The hepatic morphology of the piglets in the CON group was intact and normal (Figure 1). The hepatocytes were neatly arranged, no inflammatory cell infiltration was seen, and the stained-nuclei were more clear. However, the hepatocytes of the piglets in the DQ-CON group were loosely arranged and the phenomenon of cytoplasmic vacuole was severe and accompanied by a degree of parenchymal tissue. The phenomenon of the vacuolization of hepatocytes in the DQ-PIC piglets was more significantly alleviated than that of the piglets in the DQ-CON group.

Moreover, injecting with DQ increased the percentage of TUNEL-positive cells when compared with the CON group (*p* < 0.05, Table 4, Figure 2), whereas oral administration of PIC reduced the percentage of TUNEL-positive cells when compared with the DQ-CON group (*p* < 0.05). In addition, treating with DQ significantly decreased the protein expression level of Bcl-2 (*p* < 0.05, Figure 3) and the ratio of Bcl-2/Bax (*p* < 0.05), whereas DQ increased the Bax expression level in the DQ-CON group (*p* < 0.05) when compared with the CON group. In contrast to the DQ-CON group, treating with PIC increased the level of Bcl-2 (*p* < 0.05) and the ratio of Bcl-2/Bax (*p* < 0.05) but decreased the level of Bax in the DQ-PIC group (*p* < 0.05).

### 3.3. PIC Improves the Redox Status in the Liver of DQ-Induced Piglets

The activities of hepatic SOD and GPx were significantly inhibited in the DQ-CON group compared with the CON group (*p* < 0.05, Table 5). Conversely, treating with PIC improved the activities of hepatic SOD (*p* < 0.05) and GPx (*p* < 0.05) when compared to the DQ-CON group. Additionally, DQ reduced the amount of GSH (*p* < 0.05) and increased the amount of MDA (*p* < 0.05) in the liver compared to the CON group. In contrast, PIC increased the amount of GSH (*p* < 0.05) and decreased the amount of MDA (*p* < 0.05) in the liver compared with the DQ-CON group.

DQ reduced the gene abundance of nicotinamide adenine dinucleotide (phosphate) dependency quinone oxidoreductase 1 (*NQO1*), *SOD1* and glutathione S-transferase alpha 1 (*GSTA1*) compared with the CON group (*p* < 0.05, Table 6), but, conversely, the levels of these genes in the DQ-PIC piglets significantly increased compared with the DQ-CON piglets (*p* < 0.05). Neither DQ nor PIC altered the mRNA abundance of superoxide dismutase 2 (*SOD2*) (*p* > 0.05).

We investigated the protein expression of Keap1 and nuclear Nrf2. Keap1 protein is located outside the nucleus with beta-actin being used as the housekeeping protein while nuclear Nrf2 protein is located inside the nucleus with Histone H3 being used as the nuclear housekeeping protein. The DQ-CON piglets showed a significantly lower protein expression of SOD2 compared to the CON piglets (*p* < 0.05, Figure 4) but not compared to the DQ-PIC piglets (*p* < 0.05). However, DQ did not affect the protein expression of Keap1 and nuclear Nrf2 (*p* > 0.05), whereas PIC up-regulated the protein expression level of nuclear Nrf2 (*p* < 0.05) and decreased the level of protein Keap1 (*p* < 0.05).

### 3.4. PIC Relieves Hepatic Mitochondrial Dysfunction Caused by DQ

DQ decreased the gene abundance of *SIRT1*, PPARG coactivator 1 alpha (*PGC1α*), and mitochondrial transcription factor A (*TFAM*); the amount of mtDNA, NAD^+^, and ATP; and the protein expression level of SIRT1, while it also inhibited the activities of mitochondrial complex I, II, III, and V and CS when compared with the CON group (*p* < 0.05; Table 7, Table 8, Figure 5). The administration of PIC was effective in increasing the expression levels of *SIRT1*, *PGC1α*, and *TFAM*; the amount of mtDNA, NAD^+^, and ATP; and the protein expression level of SIRT1. PIC also improved the activities of mitochondrial complex I, II, III, and V and CS (*p* < 0.05). Neither DQ nor PIC had a significant effect on mitochondrial complex IV and the amount of NADH (*p* > 0.05).

## 4. Discussion

DQ induced a sharp decline in the body weight of the piglets during the 24 h post-injection. Images of H&E-stained liver samples prove that DQ caused obvious injury to the livers of piglets in this experiment. Elevated levels of plasma ALT and AST, an increase in the percentage of TUNEL-positive cells, and abnormal expression levels of apoptotic protein further confirm this. DQ can rapidly produce superoxide anions through the activation of molecular oxygen, which, in turn, disrupts the redox equilibrium and causes damage [19,20,21].

Moreover, in this study, hepatic mitochondrial dysfunction caused by DQ was revealed in the reduced activities of mitochondrial complex I, II, III, and V and CS, the low abundance of mitochondrial biosynthesis and function-related genes (*SIRT1*, *PGC1α*, and *TFAM*); and the reduction of the amount of mtDNA, ATP, and NAD^+^. Previous researchers have found that DQ causes mitochondrial dysfunction by attacking mitochondrial complexes, leading to the destruction of oxidative phosphorylation efficiency and increased electron leakage [26,27,28]. Mitochondria produce ATP for cell metabolism through a respiratory chain, which also acts as the electron transport chain, becoming the main source of ROS generation. Accordingly, the mitochondria are the most vulnerable to ROS [29]. Therefore, DQ might increase oxidative stress by causing mitochondrial dysfunction and, eventually, liver damage.

In the current study, PIC supplementation was shown to have the potential to inhibit the lipid peroxidation of the liver due to DQ exposure. PIC is a natural antioxidant that can directly scavenge free radicals. Previous studies have reported that PIC can effectively scavenge peroxyl radicals [25,30]. Similarly, Li et al. [31] also found that PIC showed strong antioxidant activity in scavenging hydroxyl radicals. Due to its four phenolic hydroxyl groups on the benzene ring, PIC is considered a potent antioxidant.

PIC can also stimulate the antioxidant system and enhance the antioxidant level, thereby reducing the damage of oxidative stress. In this study, we found that PIC increased the activities of antioxidant enzymes (SOD, GPx), increased the amount of GSH, and enhanced the expression levels of antioxidant-related genes (SOD1, NQO1, GSTA1), thus decreasing the amount of MDA. SOD is believed to be an important detoxification enzyme family that includes SOD1 (Cu/Zn-SOD) and SOD2 (Mn-SOD). SOD is in all oxygen-metabolizing organisms and scavenges ROS and hydrogen peroxide together with GPx and CAT using GSH as the substrate [32,33,34]. MDA is currently recognized as a sensitive indicator of lipid peroxidation that arises from a direct free radical attack on the fatty acid components of membrane lipids [35]. Similarly, Zhang et al. [36] found that PIC can significantly reduce MDA content and increase SOD and CAT activity in senile hippocampal and cortical tissues, thereby promoting their proliferation process. Volkan et al. [8] reported that PIC pre-administration can improve SOD activity and reduce lipid peroxidation. Therefore, PIC can decrease the degree of oxidative injury by directly scavenging free radicals, effectively improving the activities of antioxidant enzymes as well as the amount of GSH.

The mechanism by which PIC improved the hepatic redox status of the DQ-treated piglets may be associated with its activation of Nrf2 signals. In our experiment, PIC reversed the suppression of gene abundance of *NQO1* and *SOD1* caused by DQ and improved the nuclear translocation of Nrf2. PIC also decreased the protein expression level of Keap1. The activation of antioxidant genes, including *SOD* and *NQO1* transcription, is mediated primarily by Nrf2, which is regulated in part by the actin-associated Keap1 protein [37]. Additionally, the Nrf2 pathway can be activated by phosphatidylinositol-3 kinase (P13K)/protein kinase B (Akt), mitogen-activated protein kinase (MAPK), and protein kinase C [38]. Moreover, Zhang et al. [36] found that PIC can activate Nrf2 nuclear translocation and improved NQO1 expression in mice with aging induced by d-gal. Similarly, Lu et al. [10] found that PIC can protect APRE cells from oxidative damage induced by vitamin A via the Nrf2 pathway, which was likely completed by the activation of Akt signaling. Consequently, PIC might activate the Nrf2 pathway by promoting nuclear translocation of Nrf2 or by the activation of Akt signaling, and then by activating the expression of antioxidant genes mediated by Nrf2.

The mitochondrion is a critical organelle that participates in cellular bioenergetics, redox metabolism, and apoptosis and is sensitive to oxidative stress [29]. In the current study, PIC restored the inhibition on the activities of mitochondrial complex I, II, III, V, and CS and on ATP production. In addition, the gene abundance of *SIRT1*, *PGC1α*, *TFAM,* and mtDNA and the protein expression level of SIRT1 were improved by the administration of PIC. Mitochondrial complex I, also known as NADH dehydrogenase, can catalyze the transfer of a pair of electrons from NADH to ubiquinone. In addition, the mitochondrial complex V is considered closely related to the production of ATP, affecting the mitochondrial function [39]. *PGC1α* and *SIRT1* are considered important inducers of mitochondrial biogenesis that regulate the transcription of nuclear-encoded mitochondrial genes by interacting with *TFAM* [40]. Mitochondrial DNA is the genetic material of the mitochondria itself that can control mitochondrial biosynthesis with nuclear genes and encode proteins related to the mitochondrial respiratory chain [41]. Consequently, PIC may exert its protective effect on the mitochondria and on ATP production by improving mitochondrial biosynthesis, increasing energy metabolism efficiency, and promoting *SIRT1* expression.

In addition to reducing oxidative damage and protecting mitochondria to reduce apoptosis, PIC may directly affect the apoptotic-related pathway to exert its anti-apoptotic effect. We found that PIC promoted the protein expression of Bcl-2 while inhibited the level of Bax. Bcl-2 and Bax control the release of activated cytochrome c from the mitochondria into the cytoplasm, thereby activating caspase-3 and starting the apoptotic program. Kim et al. [42] demonstrated that PIC may reduce the apoptosis caused by oxidative stress by inhibiting the mitochondrial-mediated, caspase-dependent signaling pathway. Similarly, Zheng et al. [43] showed that PIC activated the novel PI3K/Akt/Bad signaling pathway, which regulates the activities of Bcl-2 and Bax, to exert its protective effect against Aβ-induced apoptosis in PC12 cells.

Collectively, our findings provide important evidence that PIC can protect the livers of piglets from oxidative insults caused by DQ by improving redox status, preserving mitochondrial function, and preventing excessive apoptosis. Based on previous in vitro studies on piceatannol, we innovatively used oxidative stress weaned piglets as an animal model, conducted in vivo experiments and obtained promising results. Piglet is evaluated as a model animal suitable for studying human infants owing to the similarities to human beings in anatomy and nutritional physiology, which is widely used when studying human infant digestion, organ function, growth and development and even some certain pathological conditions [44,45,46,47]. Therefore, our research on the effects of PIC on the oxidative stress piglet model may have reference value for the studies of human infants. Furthermore, this study provides new research routes for piceatannol in relieving oxidative damage of newborn animals and even human infants, and makes it a potential antioxidant feed additive, food additive or even medicine.

## 5. Conclusions

Diquat treatment induced increased apoptosis and compromised redox homeostasis of piglets. Piceatannol can protect the livers of piglets from oxidative insults caused by diquat by improving redox status, preserving mitochondrial function, and preventing excessive apoptosis.

## Figures and Tables

**Figure 1 animals-10-01239-f001:**
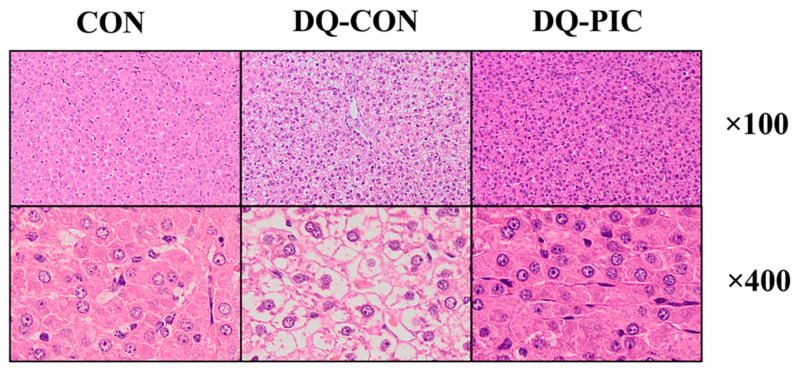
Representative images of hepatic tissue assessed by hematoxylin and eosin staining. CON, piglets were orally administrated vehicle solution (0.5% sodium carboxymethyl cellulose) and challenged with sterile saline; DQ-CON, piglets were orally administrated vehicle solution and challenged with diquat (10 mg/kg body weight); DQ-PIC, piglets were orally administrated piceatannol (80 mg/kg/day) and challenged with diquat (10 mg/kg body weight).

**Figure 2 animals-10-01239-f002:**
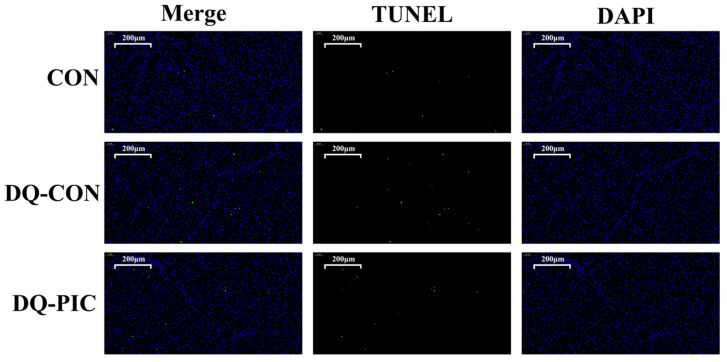
Effects of piceatannol supplementation on apoptosis of hepatic cells in diquat-induced piglets. *TUNEL*, terminal deoxyribonucleotidyl transferase-mediated deoxyuridine triphosphate nick end labeling; *CON*, piglets were orally administrated vehicle solution (0.5% sodium carboxymethyl cellulose) and challenged with sterile saline; DQ-CON, piglets were orally administrated vehicle solution and challenged with diquat (10 mg/kg body weight); DQ-PIC, piglets were orally administrated piceatannol (80 mg/kg/day) and challenged with diquat (10 mg/kg body weight).

**Figure 3 animals-10-01239-f003:**
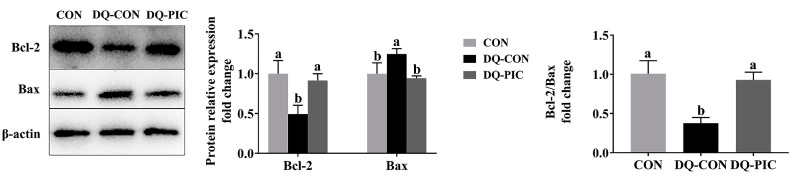
Effects of piceatannol supplementation on expression levels of protein related to apoptosis in the liver of diquat-induced piglets. The column and its bar represented the means value and SE (*n* = 6), respectively. Bcl-2, B-cell lymphoma-2; Bax, Bcl2-associated x; CON, piglets were orally administrated vehicle solution (0.5% sodium carboxymethyl cellulose) and challenged with sterile saline; DQ-CON, piglets were orally administrated vehicle solution and challenged with diquat (10 mg/kg body weight); DQ-PIC, piglets were orally administrated piceatannol (80 mg/kg/day) and challenged with diquat (10 mg/kg body weight). ^a, b^ Different letters on the shoulder mark indicate significant differences (*p* < 0.05).

**Figure 4 animals-10-01239-f004:**
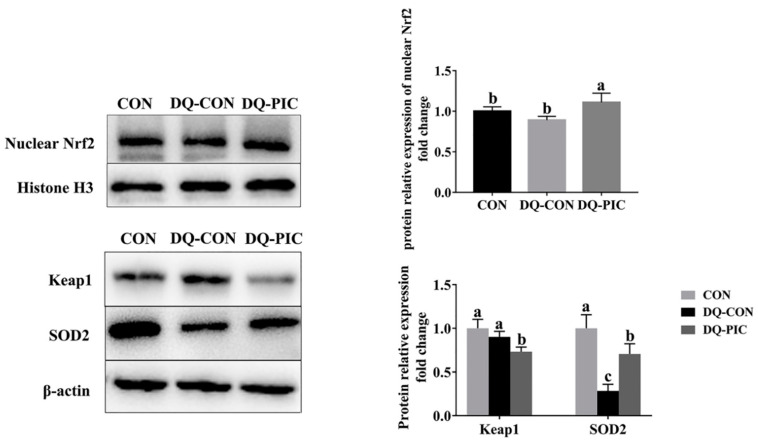
Effects of piceatannol supplementation on the expression levels of protein related to antioxidant in the liver of diquat-induced piglets. The column and its bar represented the means value and SE (*n* = 6), respectively. Nrf2, nuclear-factor-erythroid-2-related factor 2; Keap1, kelch like ECH associated protein 1; SOD2, superoxide dismutase 2; CON, piglets were orally administrated vehicle solution (0.5% sodium carboxymethyl cellulose) and challenged with sterile saline; DQ-CON, piglets were orally administrated vehicle solution and challenged with diquat (10 mg/kg body weight); DQ-PIC, piglets were orally administrated piceatannol (80 mg/kg/day) and challenged with diquat (10 mg/kg body weight). ^a, b, c^ Different letters on the shoulder mark indicate significant differences (*p* < 0.05).

**Figure 5 animals-10-01239-f005:**
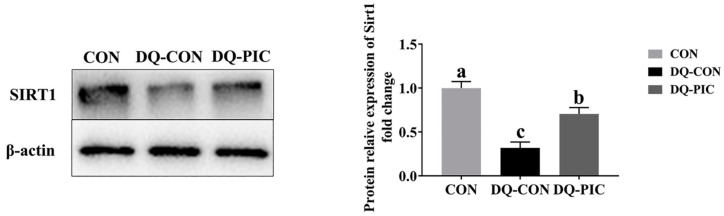
Effects of piceatannol supplementation on expression levels of SIRT1 protein in the liver of diquat-induced piglets. The column and its bar represented the means value and SE (*n* = 6), respectively. SIRT1, sirtuin 1; CON, piglets were orally administrated vehicle solution (0.5% sodium carboxymethyl cellulose) and challenged with sterile saline; DQ-CON, piglets were orally administrated vehicle solution and challenged with diquat (10 mg/kg body weight); *DQ-**PIC*, piglets were orally administrated piceatannol (80 mg/kg/day) and challenged with diquat (10 mg/kg body weight). ^a, b, c^ Different letters on the shoulder mark indicate significant differences (*p* < 0.05).

**Table 1 animals-10-01239-t001:** Composition and nutrient levels of the diet (%, as-fed basis unless otherwise stated).

Items	Contents
Maize	62.78
Soybean meal	15.00
Fermented soybean meal	7.00
Extruded soybean	7.00
Soy protein isolate	1.30
Soyabean oil	2.00
CaHPO_4_	1.80
Limestone	0.80
Salt	0.35
L-lysine-HCl, 78%	0.52
L-methionine	0.13
L-threonine	0.15
L-isoleucine	0.10
L-tryptophan	0.01
L-histidine	0.01
Calcium propionate, 50%	0.05
Premix ^a^	1.00
Total	100.00
Calculated nutrient levels	
Digestible energy, Mcal/kg	3.47
Metabolizable energy, Mcal/kg	3.30
Crude protein	20.38
SID ^b^-lysine	1.34
SID-methionine	0.40
SID-methionine + cystine	0.72
SID-threonine	0.80
SID-tryptophan	0.35
Calcium	0.82
Total phosphorus	0.43
Analyzed nutrient levels ^c^	
Digestible energy, Mcal/kg	3.47
Metabolizable energy, Mcal/kg	3.30
Crude protein	20.36
Total lysine ^d^	1.51
Total methionine	0.46
Total methionine + cystine	0.86
Total threonine	0.94
Total tryptophan	0.40
Total histidine	0.77
Total isoleucine	0.79
Total valine	1.20
Total calcium	0.82
Total phosphorus	0.65

^a^ Provide the following per kg complete diet: Vitamin A, 8000 IU; Vitamin D_3_, 3000 IU; Vitamin E, 20 IU; Vitamin K_3_, 3 mg; Vitamin B_1_, 2 mg; Vitamin B_2_, 5 mg; Vitamin B_6_, 7 mg; Vitamin B_12_, 0.02 mg; Niacin, 30 mg; Pantothenic acid, 15 mg; Folic acid, 0.3 mg; Biotin, 0.08 mg; Choline chloride, 500 mg; Fe (from ferrous sulfate), 110 mg; Cu (from copper sulfate), 7 mg; Mn (from manganese sulfate), 5 mg; Zn (from zinc sulfate), 110 mg; I (from calcium iodate), 0.3 mg; Se (from sodium selenite), 0.3 mg. ^b^ SID = standard ileal digestible; ^c^ All nutrient levels were analyzed values, except digestible energy and metabolizable energy; ^d^ The amino acid values were analyzed according to B. J. Kerr et al. [18].

**Table 2 animals-10-01239-t002:** Primer sequence used in this experiment.

Gene ^a^	Primer Sequence ^b^, 5′–3′	Accession no. ^c^	Length, bp
NQO1	F: CATGGCGGTCAGAAAAGCAC	NM_001159613.1	135
	R: ATGGCATACAGGTCCGACAC		
SOD1	F: AAGGCCGTGTGTGTGCTGAA	NM_001190422.1	118
	R: GATCACCTTCAGCCAGTCCTTT		
SOD2	F:GGCCTACGTGAACAACCTGA	NM_214127.2	126
	R:TGATTGATGTGGCCTCCACC		
GSTA1	F: ACACCCAGGACCAATCTTCTG	NM_214389.2	199
	R: AGTCTCAGGTACATTCCGGG		
SIRT 1	F: AGTTGAAAGATGGCGGACGA	NM_001145750.2	127
	R: CTCTCCGCGGTTTCTTGCG		
PGC1α	F: TGTGCAACCAGGACTCTGTA	NM_213963.2	152
	R: CCACTTGAGTCCACCCAGAAA		
TFAM	F: TGCTTTGTCTACGGGTGCAA	NM_001130211.1	100
	R: ACTTCCACAAACCGCACAGA		
mt D-loop	F: GATCGTACATAGCACATATCATGTC	AF276923	198
	R: GGTCCTGAAGTAAGAACCAGATG		
Nuclear-encoded β-actin	F: CCCCTCCTCTCTTGCCTCTC	DQ452569	74
	R:AAAAGTCCTAGGAAAATGGCAGAAG		
β-actin	F: TGGAACGGTGAAGGTGACAG	XM_003124280.5	176
	R: CTTTTGGGAAGGCAGGGACT		

^a^ NQO1, nicotinamide adenine dinucleotide (phosphate) dependency quinone oxidoreductase 1; SOD1, superoxide dismutase 1; SOD2, superoxide dismutase 2; GSTA1, glutathione S-transferase alpha 1; SIRT1, sirtuin 1; PGC1α, PPARG coactivator 1 alph; TFAM, mitochondrial transcription factor A; D-loop, mitochondrial displacement loop region, β-actin, beta actin. ^b^ F, forward primer; R, reverse primer. ^c^ Gene bank Accession Number.

**Table 3 animals-10-01239-t003:** Effects of piceatannol supplementation on average daily feed intake (ADFI), average daily gain (ADG) and feed conversion ratio (FCR) of diquat included piglets.

Items	CON	DQ-CON	DQ-PIC	SEM
28–35 days of age				
ADFI (g/d) ^1^	284.44	277.42	289.25	9.52
ADG (g/d) ^2^	214.29	211.90	226.19	9.19
FCR (g/g) ^3^	1.35	1.32	1.28	0.03
35–36 days of age				
BW change (kg/kg) ^4^	1.07 ^a^	0.91 ^c^	0.95 ^b^	0.02

^1^ ADFI, average daily feed intake; ^2^ ADG, average daily gain; ^3^ FCR, feed conversion ratio; ^4^ BW, body weight. CON, piglets were orally administrated vehicle solution (0.5% sodium carboxymethyl cellulose) and challenged with sterile saline; DQ-CON, piglets were orally administrated vehicle solution and challenged with diquat (10 mg/kg body weight); DQ-PIC, piglets were orally administrated piceatannol (80 mg/kg/day) and challenged with diquat (10 mg/kg body weight). ^a, b, c^ Different letters on the shoulder mark indicate significant differences (*p* < 0.05). Data are presented as mean (n = 6).

**Table 4 animals-10-01239-t004:** Effects of piceatannol supplementation on the contents of plasma AST and ALT and hepatic apoptosis rate in diquat-induced piglets.

Items ^1^	CON	DQ-CON	DQ-PIC
Plasma			
AST (U/L)	105.50 ± 6.98 ^c^	359.83 ± 16.44 ^a^	140.67 ± 13.33 ^b^
ALT (U/L)	49.17 ± 5.34 ^b^	257.50 ± 19.66 ^a^	57.33 ± 8.41 ^b^
Liver:			
TUNEL-positive cell percentage (%)	1.89 ± 0.070 ^b^	5.85 ± 0.077 ^a^	2.22 ± 0.086 ^b^

^1^ ALT, alanine aminotransferase; AST, aspartate aminotransferase; TUNEL, terminal deoxyribonucleotidyl transferase-mediated deoxyuridine triphosphate nick end labeling; CON, piglets were orally administrated vehicle solution (0.5% sodium carboxymethyl cellulose) and challenged with sterile saline; DQ-CON, piglets were orally administrated vehicle solution and challenged with diquat (10 mg/kg body weight); DQ-PIC, piglets were orally administrated piceatannol (80 mg/kg/day) and challenged with diquat (10 mg/kg body weight). ^a, b^ Different letters on the shoulder mark indicate significant differences (*p* < 0.05). Data are presented as mean ± SE (*n* = 6).

**Table 5 animals-10-01239-t005:** Effects of piceatannol supplementation on the redox status in the liver of diquat-induced piglets.

Items ^1^	CON	DQ-CON	DQ-PIC
SOD activity (U/mg protein)	125.61 ± 2.99 ^a^	106.43 ± 1.13 ^b^	119.88 ± 2.47 ^a^
GPx activity (U/mg protein)	147.28 ± 4.60 ^a^	116.73 ± 4.96 ^b^	136.08 ± 2.81 ^a^
GSH content (µmol/g protein)	20.21 ± 1.40 ^a^	9.54 ± 0.53 ^c^	15.75 ± 1.01 ^b^
MDA content (nmol/mg protein)	3.74 ± 0.14 ^a^	5.81 ± 0.17 ^c^	4.48 ± 0.16 ^b^

^1^ SOD, superoxide dismutase; Gpx, glutathione peroxidase; GSH, reduced glutathione; MDA, malondialdehyde; CON, piglets were orally administrated vehicle solution (0.5% sodium carboxymethyl cellulose) and challenged with sterile saline; DQ-CON, piglets were orally administrated vehicle solution and challenged with diquat (10 mg/kg body weight); DQ-PIC, piglets were orally administrated piceatannol (80 mg/kg/day) and challenged with diquat (10 mg/kg body weight). ^a, b, c^ Different letters on the shoulder mark indicate significant differences (*p* < 0.05). Data are presented as mean ± SE (*n* = 6).

**Table 6 animals-10-01239-t006:** Effects of piceatannol supplementation on expression levels of antioxidant genes in the liver of diquat-induced piglets.

Items ^1^	CON	DQ-CON	DQ-PIC
NQO1	1.00 ± 0.09 ^a^	0.64 ± 0.072 ^b^	0.97 ± 0.082 ^a^
SOD1	1.00 ± 0.05 ^a^	0.67 ± 0.047 ^b^	0.97 ± 0.041 ^a^
SOD2	1.00 ± 0.047	1.03 ± 0.107	1.00 ± 0.09
GSTA1	1.00 ± 0.091 ^a^	0.72 ± 0.045 ^b^	0.95 ± 0.077 ^a^

^1^ NQO1, nicotinamide adenine dinucleotide (phosphate) dependency quinone oxidoreductase 1; SOD1, superoxide dismutase 1; SOD2, superoxide dismutase 2; GSTA1, glutathione S-transferase alpha 1; CON, piglets were orally administrated vehicle solution (0.5% sodium carboxymethyl cellulose) and challenged with sterile saline; DQ-CON, piglets were orally administrated vehicle solution and challenged with diquat (10 mg/kg body weight); DQ-PIC, piglets were orally administrated piceatannol (80 mg/kg/day) and challenged with diquat (10 mg/kg body weight). ^a, b^ Different letters on the shoulder mark indicate significant differences (*p* < 0.05). Data are presented as mean ± SE (*n* = 6).

**Table 7 animals-10-01239-t007:** Effects of piceatannol supplementation on activities of mitochondrial complex in the liver of diquat-induced piglets.

Items ^1^	CON	DQ-CON	DQ-PIC
Complex I (U/mg prot)	5.31 ± 0.1 ^a^	3.06 ± 0.3 ^b^	5.04 ± 0.31 ^a^
Complex II (U/mg prot)	5.90 ± 0.54 ^a^	4.30 ± 0.38 ^b^	5.65 ± 0.27 ^a^
Complex III (U/mg prot)	3.54 ± 0.13 ^a^	2.28 ± 0.28 ^b^	3.23 ± 0.29 ^a^
Complex IV (U/mg prot)	18.24 ± 1.34	18.39 ± 1.41	18.97 ± 0.65
Complex V (U/mg prot)	34.68 ± 1.96 ^a^	28.19 ± 1.54 ^c^	32.87 ± 0.30 ^b^
ATP (μmol/g wet weight)	11.91 ± 0.29 ^a^	8.38 ± 0.51 ^c^	10.50 ± 0.21 ^b^
CS (U/mg prot)	100.68 ± 5.74 ^a^	47.70 ± 4.10 ^c^	84.78 ± 4.96 ^b^
NAD^+^ (nmol/mg prot)	12.51 ± 1.01 ^a^	7.15 ± 0.68 ^b^	12.85 ± 0.93 ^a^
NADH (nmol/mg prot)	6.76 ± 0.72	7.80 ± 0.29	6.82 ± 0.95

^1^ ATP, adenosine triphosphate, CS, citrate synthase; NAD^+^, nicotinamide adenine dinucleotide; NADH, reduce nicotinamide adenine dinucleotide; CON, piglets were orally administrated vehicle solution (0.5% sodium carboxymethyl cellulose) and challenged with sterile saline; DQ-CON, piglets were orally administrated vehicle solution and challenged with diquat (10 mg/kg body weight); DQ-PIC, piglets were orally administrated piceatannol (80 mg/kg/day) and challenged with diquat (10 mg/kg body weight). ^a, b, c^ Different letters on the shoulder mark indicate significant differences (*p* < 0.05). Data are presented as mean ± SE (*n* = 6).

**Table 8 animals-10-01239-t008:** Effects of piceatannol supplementation on expression levels of genes related to mitochondrial biosynthesis and function in the liver of diquat-induced piglets.

Items ^1^	CON	DQ-CON	DQ-PIC
SIRT1	1.00 ± 0.096 ^a^	0.70 ± 0.033 ^b^	0.92 ± 0.043 ^a^
PGC1α	1.00 ± 0.060 ^a^	0.73 ± 0.030 ^b^	0.99 ± 0.038 ^a^
TFAM	1.00 ± 0.055 ^a^	0.64 ± 0.042 ^b^	0.96 ± 0.070 ^a^
mtDNA	1.00 ± 0.11 ^a^	0.40 ± 0.11 ^b^	1.00 ± 0.19 ^a^

^1^ SIRT1, sirtuin 1; PGC1α, PPARG coactivator 1 alpha; TFAM, mitochondrial transcription factor A; mtDNA, mitochondrial DNA; CON, piglets were orally administrated vehicle solution (0.5% sodium carboxymethyl cellulose) and challenged with sterile saline; DQ-CON, piglets were orally administrated vehicle solution and challenged with diquat (10 mg/kg body weight); DQ-PIC, piglets were orally administrated piceatannol (80 mg/kg/day) and challenged with diquat (10 mg/kg body weight). ^a, b^ Different letters on the shoulder mark indicate significant differences (*p* < 0.05). Data are presented as mean ± SE (*n* = 6).

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
