# Peer review of "Piceatannol Ameliorates Hepatic Oxidative Damage and Mitochondrial Dysfunction of Weaned Piglets Challenged with Diquat"

_animals, 2020, doi:10.3390/ani10071239_

Round 1
Reviewer 1 Report
I have only few comments:
- For repeatability of experiment is necessary to show the origin of vehicle solution, diquat and piceatannol (Company, Country).
- From Figure 2 is hard somethig to see – if possible change the graphic of Figure 2.
Reviewer 2 Report
- The article is well written and structured, with a good dose of scientific analysis, although some sections are missing as I will comment later.
- Every scientific article should have a hypothesis and initial objectives, please add them.
- In material and method section, I miss a section in which you describe the detail of PIC and DQ used in the trial, especially detail of PIC i.e. dosage, precedence of the active principle, application etc.. This section should be placed before “2.2. Animals and treatments”
- The analyzed amino acid values are shown in table 1, but the material and methods are not mentioned as analyzed. Please add them.
- In tables, please aligned the first column (Items) on the left side. In addition, use letters such as “a” , “b”, “c” or “ab” when the means are statistical different instead “* and #”
- The discussion it is okay, but I would suggest to structure it the same way that the results section (3.1 and 3.2).
Minor comments
Line 86à Add initial body weight and standard deviation of the piglets
Line 98à Replace “Feed efficiency” by “Feed conversion ratio”. Please modified it although manuscript.
Line 100à Please divided the table in 3 parts; Ingredients; Calculated analysis (add it); Analyzed analysis. In addition, I would suggested you to include digestible AA instead Total AA.
Line 107à Remove “2.3”
Table 3 à
- Remove * and # and use letters “a” , “b” or “ab” when the means are statistical different
- Remove the standard error of the means
- Use numbers for the footnote and please use different footnote for ADG, ADFI, FCR and BW. It is also possible to describe this variables in the title like that: “Effects of Piceatannol supplementation on average daily feed intake (ADFI), average daily gain (ADG) and feed conversion ratio (FCR) of diquat included piglets”.
Line 219. Rewrite “Treatment with PIC Decreases the Hepatic Injury Caused by DQ in Piglets”. Avoid to include result in the subheading (remove at least decreases).
Line 237 and 238: Full name for PIC and DQ
Reviewer 3 Report
Comments to Authors:
The authors tested the effects of piceatannol (PIC) on hepatic damage caused by diquat (DQ)-induced oxidative stress in weaned piglets. The treatment ameliorated most of the variables investigated. Specifically, 80 mg/kg/day PIC improved hepatic redox status, preserved mitochondrial function and prevented excessive apoptosis of hepatic cells.
There are some points to be fixed.
Major points are
- L21 what does mean “more “in the sentence
- In L10-14 it is reported that DQ injection is followed by PIC administration. However, in L93-95 it is reported that PIC was administered from 28 to 35 days, while DQ was injected at 35 day. Please correct this part in Simple summary (L10-14).
- In L93-94 you report the doses of DQ and PIC used in the experiment. Did you assess them after some trials with different doses? If yes, please report the data. Did you find them in previous studies? If yes, please report the references.
- The hepatic damage given by oxidative stress was assessed through different experiments and observations. Did you perform a direct measurement of intracellular ROS level? The presence of oxidative stress is very clear considering all the results reported, but a direct ROS measurement could potentiate your observations.
- In Fig.4 there are the results of Western Blot of Nrf2 and Keap1. You use H3 as housekeeping gene for Nrf2 and beta-actin as housekeeping gene for Keap1. You should report this difference in L282-285.
- In the final part of the discussion you summarize all the findings of the study. You should report which are the future perspectives of this study, such as how these results could be useful for human infants. Moreover, you should underline the fact that this study is innovative considering the previous literature. In my opinion the animal model used and the promising results are two important and innovative aspects to be considered.
- Taqble 3PIC intake is missing
Minor points are:
- L108: Please add the paragraph number 2.3 before “sample collection”
- L166: Please add the paragraph number 2.8 before “REAL-time…” and use Italic type
- L269: Please put a line break after “3.3 PIC improves…”
- L318: Please use Italic type for “3.4 PIC relieves…”
- Under all the Figures and Tables: please correct the line “Significant different…” with “Significant differences…”
Round 2
Reviewer 3 Report
After your corrections the article is more clear and complete. However, there are some minor issues to be fixed.
-Following your reply about DQ and PIC doses, I suggest you to add in the text the correlation between resveratrol and PIC that you underlined in your “Response to reviewers”. In my opinion this element clears the choice of PIC dose.
-Regarding L451-455, where you underlined the innovative aspects of the study, you added also some possible future perspectives, such as the potential use of PIC as feed/food additive. In my opinion you should show in a more detailed way which are the connections (given by the results that you obtained) between animal model (piglets) and human infants, perhaps by searching in the literature if there are previous similar works or attempts to implement these kind of results in human infants.
